# Comparison of Optimisation Algorithms for Centralised Anaerobic Co-Digestion in a Real River Basin Case Study in Catalonia

**DOI:** 10.3390/s22051857

**Published:** 2022-02-26

**Authors:** David Palma-Heredia, Marta Verdaguer, Vicenç Puig, Manuel Poch, Miquel Àngel Cugueró-Escofet

**Affiliations:** 1Laboratory of Chemical and Environmental Engineering (LEQUIA), Institute of the Environment, Universitat de Girona, MªAurelia Capmany, 69, 17003 Girona, Spain; marta.verdaguer@udg.edu (M.V.); manuel.poch@udg.edu (M.P.); 2Advanced Control Systems (SAC) Research Group, Polytechnic University of Catalonia (UPC-BarcelonaTech), Terrassa Campus, Gaia Research Bldg, Rambla Sant Nebridi, 22, 08222 Terrassa, Spain; vicenc.puig@upc.edu (V.P.); miquel.angel.cuguero@upc.edu (M.À.C.-E.); 3Institut de Robòtica i Informàtica Industrial (CSIC-UPC), 46 Llorens i Artigas Street, 08028 Barcelona, Spain

**Keywords:** anaerobic co-digestion, ant colony optimisation, particle swarm optimisation, genetic algorithms, waste management, circular economy

## Abstract

Anaerobic digestion (AnD) is a process that allows the conversion of organic waste into a source of energy such as biogas, introducing sustainability and circular economy in waste treatment. AnD is an intricate process because of multiple parameters involved, and its complexity increases when the wastes are from different types of generators. In this case, a key point to achieve good performance is optimisation methods. Currently, many tools have been developed to optimise a single AnD plant. However, the study of a network of AnD plants and multiple waste generators, all in different locations, remains unexplored. This novel approach requires the use of optimisation methodologies with the capacity to deal with a highly complex combinatorial problem. This paper proposes and compares the use of three evolutionary algorithms: ant colony optimisation (ACO), genetic algorithm (GA) and particle swarm optimisation (PSO), which are especially suited for this type of application. The algorithms successfully solve the problem, using an objective function that includes terms related to quality and logistics. Their application to a real case study in Catalonia (Spain) shows their usefulness (ACO and GA to achieve maximum biogas production and PSO for safer operation conditions) for AnD facilities.

## 1. Introduction

In the context of global climate change with rising and more extreme events—such as droughts and floods—which will likely provide growing uncertainty to water demand and jeopardise the availability of specific resources, there is a growing interest in the adaptation and use of technologies related to the circular economy that promote environmental sustainability. In this framework, resource recovery is a key issue for industrial and environmental processes and shows a wide spectrum of study possibilities. In water sanitation, wastewater treatment plants (WWTPs) offer a wide range of possibilities for resource recovery, mainly related to sludge treatment processes [1,2,3,4,5,6,7] as biogas generation via the substrate co-digestion process, which can be an alternative source for thermal and electrical energy production [8,9,10,11,12,13,14]. This potential for biogas generation could be translated as well to a source of renewable natural gas, which has specific composition requirements that demand high-tech sensors to assure its quality no matter its origin, as those developed in [15,16]. Due to their potential for resource recovery and the further implications in the water–food–energy nexus, WWTPs have been a research focus from different areas of expertise: from modelling and engineering design [17,18,19,20,21,22,23,24] to process dynamics, simulation and integration [25,26,27,28].

Anaerobic digestion (AnD), a complex process involved in biogas production, has a delicate balance of substrate composition. The optimal performance requires avoiding process inhibition and maximising biogas generation. The optimal balance may be achieved with the correct mixture of available substrates, but this task is challenging and difficult to achieve manually due to the high combinatorial possibilities and changing availability of substrates of heterogeneous nature. The complexity increases when the process is co-digestion, with the addition of residual substrates produced by agro-food and similar industries, each with its own dynamics of substrate generation and composition [29,30,31,32]. Additionally, not all WWTPs have an anaerobic digester. Therefore, optimisation also requires logistical challenges to process a maximum volume of the available substrates in a certain geographical area and its travel logistics (of sludge and co-substrates) from its origin to the destination digester. 

Hence, dealing with such complexity is a former step to tackle optimal co-digestion in a complex network composed of many substrate sources—including WWTPs without AnD processes and industrial producers—and several co-substrate receptors. These will be located in different geographical places. As a result, the logistics of substrates will be affected by the geographical distance between actors involved and the restrictions related to the receptors.

Optimisation of the individual digester feed requires optimal blending of different co-substrates in order to fulfil the volumetric and compositional requirements of the anaerobic procedure. This problem can be understood as a multidimensional knapsack problem (MKP) [33,34,35]. The MKP is an NP-hard problem [36] and has been widely studied in the literature. To solve this type of problem, the use of combinatorial optimisation metaheuristics is proposed in [37,38], mainly when a high number of restrictions are presented [39]. 

Many tools have been developed to this end, either focused on modelling, control of the optimum co-substrate blending, or system operation, as shown in [37,39,40,41,42,43]. In [37,40], identification and modelling of critical parameters are performed; in [39] control schemes based on the composition qualities are developed; and in [42,43], optimised control strategies are implemented according to blend composition. In [41], logistics are also included to optimise the performance of a single anaerobic digester with co-digestion strategies. 

However, in real-world installations, most of these systems are managed and supervised not in a single fashion but in a network fashion. Thus, proper system management requires simultaneous consideration of the entire AnD network to select which combination is the best for each digester to maximise the potential of the overall infrastructure. Besides, literature on this matter is relatively scarce due to its ad-hoc nature. There is literature related to optimised placement of new AnD plants, such as in [44], but it lacks optimisation of the operational part involved in the feeding of the anaerobic digesters. Very specific works can be found about optimisation of supply chain networks in the field of waste valorisation, such as in [45], where an integrated geographical information system (GIS)-based optimisation is performed, but it requires highly detailed and tailored data, so its implementation becomes time-consuming and highly dependent on data availability; furthermore, it does not tackle process optimisation regarding waste processing facilities. Regarding logistics, other works can be found for path planning optimisation such as in [46], where truck routes are traced based on GIS-oriented algorithms, or in [47], where a smart waste bin prototype is developed for sensor-based waste classification. As it can be seen, there is a gap in the literature regarding network optimisation of existing waste management facilities (such as AnD plants) that would include both logistics (i.e., minimising route impact and length) and quality (i.e., improving process performance) optimisation. This is a necessary gap to fulfil, since, as stated before, AnD networks are currently managed in an ad-hoc, manual fashion by the practitioners, which is dramatically time-consuming and needs highly qualified personnel. Although there are currently different approaches that could help overcoming specific parts of this challenge (i.e., those observed in [44,45,46,47]), none of them can currently successfully accomplish the overall task. 

The approach presented in this study continues the work introduced in [48] where the optimisation problem of blending in anaerobic co-digestion (AnCD) is handled by an ant colony optimisation (ACO) algorithm in a synthetic case study with realistic conditions, simulating a centralised AnD single-stage reactor that received feed once a day. In [41], the work is extended while considering the quality, social and travel logistics of the co-substrates, analysing its importance to the overall optimisation. In addition, ACO has been implemented in real-world waste sector case studies, e.g., [48,49]. Here, the work is extended to a similar real-world case study considering multiple receptors in the geographical area of the Besòs River basin (Catalonia, Spain). The data used correspond to the real operation conditions of this area. For these conditions, the authors have also evaluated the results obtained using different optimisation approaches such as ACO, genetic algorithms (GA) and particle swarm optimisation (PSO).

ACO, GA and PSO algorithms were selected as convenient approaches to tackle a problem of the nature stated here after reviewing applications of similar nature in the literature. In [50], a review of nature-inspired algorithms is performed, including GA, ACO and PSO—among others—for AnD modelling and optimisation, showing how in the field of AnD, PSO obtains better performance in substrate feed optimisation for agricultural biogas plants than other evolutionary methods. For example, in [51] genetic algorithms are used to minimize the environmental impact caused by mine water. The main drawback for PSO pointed out in [50] is premature convergence, since particles may become trapped in local optima or suffer stagnation, but this may be solved by a partial restart of the process introducing new particles in the search space. In [52], ACO and GA are applied to optimise the route of waste collection vehicles for municipal waste collection and transportation—the highest cost of the entire waste management system—with similar performance attained by both algorithms; however, only the problem of travel logistics is considered, not the blending of municipal waste. Ref. [53] proposes a nonlinear model predictive control strategy using the MATLAB BioOptim toolbox, developed by the same authors, for optimal control of substrate feed to AnD operation of an agricultural biogas plant, with a graphical user interface (GUI) integrating a fitness function including different operating constraints and parameters such as pH, solids or methane concentration, and using evolutionary optimisers such as PSO, covariance matrix adaptation evolution strategy (CMAES). Alternatively, they propose differential evolution (DE), which achieved better performance with PSO but without considering the substrate travel logistics in the optimisation. Ref. [54] presents a prediction and optimisation method using a multi-layer perceptron artificial neural network (ANN) and PSO for the maximisation of biogas generation in a real wastewater treatment facility. A similar approach is presented in [55], where modelling and optimisation of biogas production with mixed substrates are obtained with a combination of ANN and GA methods. 

Additionally, ref. [55] and references therein point out how stochastic global optimisation algorithms (SGOAs), such as PSO, the ACO, and GA, among others, are considered efficient alternatives in the design of optimal production media and optimal process operating conditions in fermentation research and can significantly reduce the process development time. Regarding the comparison of the optimisation algorithms selected here solving combinatorial optimisation (CO) problems, in [56], ACO and GA are compared, both achieving good performance but with GA exhibiting slightly better performance than ACO. In the latter reference, it is also mentioned how trimming of specific parameters for both optimisers—e.g., number of iterations, evaporation coefficient and number of ants for ACO, or chromosome population, crossover and mutation probabilities for GA—is required to achieve good performance in both cases. In [57], relationships between GA and ACO-type algorithms are detailed, presenting their similitudes and showing how they use similar principles to succeed in CO problems with globally convex structure of its solution space. Overall, SGOAs such as ACO, GA or PSO have shown good performance in the type of applications presented and hence demonstrated suitability for non-convex nonlinear multidimensional optimisation problems, as presented here. Optimal blending for AnCD is considered, e.g., in [40,55], but to the knowledge of the authors, the optimisation of such blending combined with the travel logistics of co-substrates in a centralised multi-receptor co-digestion strategy has not yet been studied. Additionally, a comparison between different suitable optimisation algorithms for such applications, i.e., ACO, GA and PSO, is presented.

The application of optimisation strategies in AnD allows a significant enhancement of co-digestion strategies [30] maximising biogas production and minimising associated risks to each AnD operation (e.g., overdosing or acidification). In this work, the performance of each optimisation approach considered is evaluated on a real case study in the area of the Besòs River basin in Catalonia, including a network of substrate generators and three anaerobic digesters. Hence, the objective of this study is to develop a tool that is able to optimise the centralised digestion process of an AnD network with multiple waste sources and waste receptors by means of three evolutionary optimisation algorithms—namely, ACO, GA and PSO. Such a tool is tested in a real case study to further analyse and compare the performance of each algorithm in the overall AnD network optimisation.

## 2. Material and Methods

### 2.1. Optimisation Algorithms Considered

The optimisation algorithms presented here fall within the set of SGOA, concretely in the subset of evolutionary algorithms (EA) for GA—which use mechanisms inspired by biological evolution, e.g., mutation or recombination to achieve the goal of optimisation—and in the subset of swarm intelligence methods for ACO and PSO based on the collective behaviour of self-organised decentralised systems, respectively. SGOA algorithms have been widely used to solve NP-hard combinatorial optimisation problems, such as that presented in this study, which deterministic optimisation methods fail to handle due to their complexity.

Regarding each proposal, ACO is a metaheuristic approach that has been shown to be effective in solving a variety of NP-hard problems [58]. The algorithm is based on simulation of the behaviour of real ants in their search for food. When ants find food, they leave a pheromone trail on their path. Then, new ants follow that trail. In this way, an increasing number of ants are concentrated in places where there is food. In a similar way, the virtual ants construct a solution moving through the graph that represents the search space of solutions. Their paths are guided by a probabilistic state transition rule, which is based on pheromone trails and specific heuristic information. The algorithmic procedure is iterative. At each iteration, the pheromone trails are updated by applying an evaporation coefficient (when the value selected is not part of a feasible solution). To avoid rapid stagnation of the solution, the ACO algorithms can use several strategies [59], such as that related to the limitation of the pheromone trails between maximum and minimum values. Max–Min Ant System [60] uses this procedure.

The GA is a metaheuristic approach also used in combinatorial optimisation problems. It is based on the mechanics of natural selection and natural genetics. GA applications cover a range of combinatorial optimisations, e.g., hydraulic model calibration [61], performance of photovoltaic systems under variable atmospheric conditions [62], and sensor placement for leak detection in water distribution networks [39,63]. The GA is based on three main parameters: selection, crossover and mutation. The population matrix is randomly generated and consists of the design variables, and the best variables are selected according to their fitness value. From these solutions, new solutions are produced via the crossover operator [64]. The mutation operator is finally employed to avoid the algorithm converging to local optima (i.e., to maintain the genetic diversity). 

The GA cycle is repeated through a number of generations until a stopping criterion is met. It is worth noting that elitism is not generally considered an operator in the canonical GA. However, it is deemed a robust and effective operator because it leads the optimisation procedure towards the optimal solution. Accordingly, this operator stops the best solutions from being mutated. In this way, the best solutions of each generation would pass to the next, unaltered. Over the course of the algorithm and through a sufficient number of generations, the traits of these solutions would transfer to their offspring, increasing the chance of producing new solutions whose fitness function values might be better than their parents [64]. Some drawbacks of GAs are noted in [61], e.g., achieving a global optimum for large and complex systems is not guaranteed, which is also a drawback for ACO. In [57], the relation between GAs and ACO is noted.

PSO is a recently developed EA that includes features such as easy implementation for solving practical problems, high accuracy and fast convergence of the solution as some of its main advantages [65,66]. While similarities exist in the iterative nature of PSO and GAs, conversely, in PSO, there are no, e.g., “crossover” or “mutation” operations. Instead, PSO is based on a population of candidate solutions, defined as particles. The set of particles composes a swarm, where each individual flows through the parameter space. The flow of such particles is defined by trajectories, which are driven by the best performance of the particle and the neighbouring particles in the parameter space. The initialization of particle swarm is random. The initial solution of each particle represents an alternative solution; that is, each particle has its own initial position and speed and is randomly distributed in each position of the feasible solution space to be searched. Therefore, the initialization of the particle swarm represents the preparation of the particle swarm search. Its size is determined by its speed and position, and the particle update is based on the comparison of the fitness values between each search particle and its neighbouring search particles to determine the necessity of updating a particle. The updated particle adjusts its speed and position according to the particle’s new flight path model, which is based on the best results achieved by its neighbouring particles. These conditions yield different optimal experiences for different particle subgroups, which dynamically evolve according to the current position of particles, the particle current velocities, the distance between each particle of the subgroup and its best position and the distance between each particle of the subgroup and the best position of the whole subgroup. 

The PSO algorithm does not need cross-mutation or other genetically inspired operations, so the algorithm has fewer parameters and is still high efficiency [65]. These properties are suitable for both engineering applications and scientific research, and a significant number of research results have been produced in recent years [67]. For example, in [68], PSO is applied for function optimisation regarding eco-economics modelling and assessment; in [69], it is used as part of fuzzy systems developed to optimise fuel consumption of hybrid vehicles; and in [70], PSO is used to train neural network models and perform real-time optimisation.

### 2.2. Centralised Co-Digestion as an Optimisation Problem

Mathematical optimisation involves the selection of one solution amongst a set, according to some criterion and constraints (that is, the optimisation problem). This optimisation problem can be stated as:(1)$$\underset{x\in X}{\min}f\left(x\right)\;\mathrm{subject}\;\mathrm{to}\;g\left(x\right),$$
where *f*(*x*) is the objective or cost function, *X* is a feasible region and *g*(*x*) are the constraints that have to hold to find a minimiser *x** of *f*(*x*) such that $f\left(x^{*}\right)=\min_{x\in X}\;f\left(x\right)$. ACO, GA and PSO introduced in Section 2.1 are algorithms aimed at finding the optimal solution of the optimisation problem posed in (1)—i.e., minimise the objective function *f*(*·*) subject to the set of constraints *g*(*x*) that apply—which here consists of the selection of the best substrates and volumes according to a set of restrictions related to the operation of the anaerobic digester. In addition, the cost function allows quantifying each alternative potential solution according to (1), involving the calculation of a value or “cost” associated with each alternative considered to find the optimal solution.

The problem statement is similar to that presented in [41], although the number of waste receptors increases from one to three, thus making necessary a reformulation of the optimisation problem involved. Specifically, it is required to increase the dimensions of all data vectors that define each generator-receptor interaction and their subsequent calculations. In addition, matrix operations are repeated per new dimension (i.e., waste receptor) added.

This optimisation problem, which can be understood as a MKP and is of combinatorial nature, can be represented as a matching problem. It is defined with a graph *G =* (*N, E*) that summarises all the possible combinations. The graph consists of *N* vertices (or nodes) and *E* edges or pairs of vertices. Specifically, for the case of AnD optimisation, a bipartite graph can be used to represent the posed optimisation problem to differentiate between the set of waste generators (*N*_1_), containing *W* nodes, and the set of waste receptors (*N*_2_), containing *R* nodes. For the specific optimisation problem, all *W* nodes are connected to each *R* node, thus resulting in a total of *W·R = E* edges. Figure 1 shows a generic representation of the defined matching problem applied to AnD co-digestion optimisation.

A set of substrate generators w∈{1,…,N} is considered. The volume of each substrate $V_{w}$ can be selected as a contribution to any of the AnD systems. The binary decision variable $y_{w}^{s}$ allows generating array volumetric possibilities ($V_{w}^{s}$, with $s\in \left\{0\ldots ,l_{w}\right\}$ that are determined as a multiple of a number (e.g., 1000 by default) such that $1000l_{w}=V_{w}$. The selection of each volumetric possibility is determined by the corresponding value of the binary decision variable, $y_{w}^{s}$, where y∈{0,1}, with $y_{w}^{s}=0$ when the corresponding volumetric configuration is not selected, and $y_{w}^{s}=1$ when it is selected. Note that for each waste generator *w*, there are $l_{w}$ different volumetric configurations in $y_{w}^{s}$, but only one is selected at a time, i.e., $\sum_{s=1}^{l_{w}}y_{w}^{s}=1\;\forall w\in \left\{1\ldots ,N\right\}$.

To normalise the objective function, selected volumes $V_{w}^{s}$ are divided by the maximum volume from their corresponding waste generator ($V_{w}$). This approach provides values of the cost function between 0 and 1, where the closest to 1 the better the solution. However, note that the ACO algorithm looks for a maximum of the objective function, while GA and PSO look for a minimum. This behaviour is considered using the constant K∈{−1,1}, which depends on the algorithm considered: for the ACO algorithm K=1 and for the GA and PSO algorithms K=−1. Hence, cost index *B* would take positive values between 0 and 1 for ACO and negative values between 0 and −1 for GA and PSO. Instead, the absolute value is taken for all three algorithms.

$F_{w}^{c}$ (c=1,…, 3) and $T_{w}$ are the set of dimensionless coefficients corresponding to the substrate characterisation and the quality term $\left(\sum_{c=1}^{3}F_{w}^{c}\right)\rho_{q}$, already used and explained previously in [41,46] and defined as shown in Figure 2.

$F_{w}^{1}$ is a coefficient related to the potential biogas production, measured as a function of the Chemical Oxygen Demand (*COD*) content. $F_{w}^{2}$ indicates the ratio of *COD*/*TN* (where *TN* refers to Total Nitrogen), a useful measure to prevent situations of acidification and other undesired reactions of the AnD process, as long as it is maintained around the range of 20–60. $F_{w}^{3}$ is linked to the alkalinity (Alk) concentration, and it is associated with a restriction ranging from 2500 to 6000 mg CaCO_3_/L integrated within all optimisation algorithms. $T_{w}$ is a coefficient of the utmost importance since it describes the toxicity level of all waste fluxes, which should be kept at the lowest level possible (specifically below 2.1 mg Pb/L).

The *N* different substrate generators are located at different distances ($d_{w}$) from each anaerobic digester. The conveyance of the selected volumes implies a travel distance $\;d_{w}$ (in km) with an economic cost $x_{w}$ (in €/km) and a social impact $I_{w}=1,\ldots ,\;3$ (dimensionless). The higher the value of $I_{w}$, the higher the social impact of the related route (e.g., proximity to sensitive areas due to pollution, traffic density, or pedestrian presence). Since each route is different for each generator, different values are assigned to approach the logistic impact of the corresponding waste generator, so a value for $I_{w}$ is assigned for each sludge/substrate generator depending on its route to the ST.

The coefficient weight $\rho_{q}$ (dimensionless) is related to the quality term $\left(\sum_{c=1}^{3}F_{w}^{c}\right)\rho_{q}$, and the coefficient $\rho_{x}$ (dimensionless) is the coefficient that weights the logistics term $\frac{\rho_{x}}{X_{w}d_{w}I_{w}}$. Each weight is given a value of 0.5 to provide a balance between the quality and logistics terms in the optimisation. Selected volumes of each substrate to each receptor contribute to the input to the AnD network, and the aforementioned parameters constitute the objective or cost function *f*(*x*).

Additionally, the optimisation problem presented considers a set of restrictions *g*(*x*) related to each of the total inputs to each of the receptor systems, based on those presented in [41,48]. The first restriction is the sum of accepted substrates $\sum_{w=1}^{N\;}\sum_{s=0}^{l_{w}}y_{w}^{s}V_{w}^{s}$ must not exceed the maximum acceptable volume V for each AnD system. Moreover, the *COD*/*TN* ratio, related to the dimensionless coefficient $F_{w}^{2}$, must be kept within the range $\left[C_{min}^{2},C_{max}^{2}\right]$. The alkalinity concentration, related to the dimensionless coefficient $F_{w}^{3}$, must also be kept within the range $\left[C_{min}^{3},C_{max}^{3}\right]$. The toxicity level, related to $T_{w}$, does not require restriction since the corresponding coefficient $T_{w}$ is considered to be restrictive enough, as shown in Figure 2.

In addition, an estimation of the produced biogas is made assuming a conversion factor of 0.268 m^3^ biogas/kg *COD*. Finally, note that the cost function presented in this work is adapted from [41,48], where the ACO algorithm was used for waste management optimisation in a similar fashion but limited to one AnD receptor.

The objective function *f*(*x*) for the presented optimisation problem is as follows in (2). However, note that the performance comparison of the ACO, GA and PSO algorithms is not conducted directly on the value of the optimised objective function, $B^{\prime }$, but on its absolute value, B, as shown in (3).
(2)$$B^{\prime }=K\left\{\sum_{w=1}^{N\;}\sum_{s=0}^{l_{w}}y_{w}^{s}\frac{V_{w}^{s}}{V_{w}}T_{w}\left[\left(\sum_{c=1}^{3}F_{w}^{c}\right)\rho_{q}+\frac{\rho_{x}}{X_{w}d_{w}I_{w}}\right]\right\}$$
(3)$$B=\left|B^{\prime }\right|$$

## 3. Results

### 3.1. Case Study

The case study includes a network of 19 organic waste generators and three organic waste receptors. These 22 locations (i.e., 19 generators and 3 receptors) are part of the wastewater treatment system managed by Consorci Besòs Tordera (CBT), a public local water administration composed of 64 municipalities in four different regions of Catalonia (Spain) with a population of approximately 470,000 inhabitants. This case study and its anaerobic network system were also considered in [41]. Figure 3 shows the corresponding bipartite graph of the case study.

The three organic waste receptors (R1–R3, or nodes 1–3 of Figure 3) refer to three separate WWTPs that produce their own sewage sludge, but that also have available AnD technology. Due to oversized design, which is a usual practice in WWTP design [71], these AnD systems in R1–R3 have available capacity. This free excess capacity can be used to accept wastes from external sources, such as the undigested sewage sludge of W1–W12 or the industrial substrates from C1–C7.

The 19 waste generators consist of 12 WWTPs that produce undigested sewage sludge (W1–W12, or nodes 4–15 of Figure 3) and seven industrial substrate generators (C1–C7, or nodes 16–22 of Figure 3), which were considered suitable sources of organic waste for the AnD network under study. Each of these locations is a separate and independent system that must manage its own waste produced as best as possible. Additionally, seven industrial substrate generators have been previously verified as feasible substrates for AnD by CBT technical services.

### 3.2. Simulation Methodology

The algorithms used in this work have been implemented in the MATLAB environment. Simulations were performed with a Lenovo ThinkPad (Lenovo Group, Ltd., Girona, Spain) L14 Gen1-20U10016SP ×64 using the OS Microsoft Windows 10 Pro and an Intel(R) Core(TM) i7-10510U CPU processor (1.80 GHz, 2304 MHz) consisting of four main processors and eight logic processors.

The main optimisation parameters of both GA and PSO algorithms were trimmed in an attempt to select the most suitable array to provide reliable results. Accordingly, the same procedure was already performed for the ACO algorithm to determine the values of its corresponding optimisation parameters in [48], where the same optimisation parameters are used in this work.

For the ACO algorithm, an initial population of 100 individuals (or ants) and 500 iterations per repetition is set, and the values used for the algorithm optimisation parameters are α=1, β=2 and ρ=0.98, each corresponding to the importance assigned to the pheromone trail, the importance assigned to the heuristic information and the persistence degree or pheromone evaporation, as explained in [38,58,60]. For GA, the initial population is set to 100, the total number of iterations (or generations) is set to 500, the crossover fraction is set to 0.8 and the fraction of elite children is set to 5% of the corresponding total children. For PSO, the initial population was set to 100, the total number of iterations was set to 500, cognitive attraction was set to 0.8 and the social attraction factor is set to 1.25. Table 1 and Table 2 summarise trimming tests for GA and PSO, respectively, where the best results are obtained for higher values of objective index *B*.

For the sake of performance comparison, some parameters were fixed for the three algorithms. The fixed parameters are the number of independent simulations (set to 10, the best result is selected), the population (set to 100 individuals), and the maximum number of iterations (set to 500). With these constraints on algorithm trimming, a performance comparison of ACO, GA and PSO was conducted.

The comparison of ACO, GA and PSO performances is based on the value of the fitness function, execution time and an array of technical variables related to the total expected performance of the optimised AnD network: total daily biogas production (in Nm^3^), average organic load (in kg of COD per m^3^ of volume of the digestion system and day), average carbon to nitrogen ratio (C/N), and average alkalinity (in mg of CaCO_3_). All algorithms are tested with data from a real case study as the main simulation scenario. However, other synthetic scenarios are tested to further compare the performance of each algorithm under different scenario conditions.

In the approach presented here, simulated scenarios are based on the waste generator data in Table 3, alongside route distance and receptor system characterisation. For all the 19 waste generators (i.e., the 12 WWTPs without AnD and the seven substrate generators), the addition to the AnD network is optimised. For each of the three AnD systems (i.e., receptors R1, R2 and R3), different volume constraints have been determined, according to operational data and assuming a limit to the hydraulic retention time of 20 days (below that retention time, AnD efficiency is expected to greatly decrease).

Each simulation for ACO, GA and PSO is repeated 10 times since these algorithms have probabilistic, iterative-based search methods. The best solution among these runs is selected for further analysis, although the average fitness function is also registered for discussion.

The data obtained for ACO, GA and PSO (each comprising 10 repetitions of the corresponding algorithm) are compared for every simulated scenario. The baseline scenario (i.e., Scenario 0) corresponds to the real case study, as described in Table 3. Additional synthetic Scenarios 1–4 are simulated, and their corresponding data are created from alterations of the baseline scenario, as described in Table 4.

While for Scenarios 1 and 2 any modification is viable, for Scenario 3, an increase in volume involves a significant increase in execution time. This is because the optimisation problem works around combinations of fixed volumes, and an increase in volume would involve a higher number of possible combinations for the algorithms to consider (i.e., an increase in the search space), hence the expected increase in execution time. Thus, volume modification for Scenario 3 was limited to a triple increase in the baseline scenario volume. Alternatively, for Scenario 4, the C/N ratios were modified while being kept below 60 to facilitate the algorithms in finding a viable solution. This measure was adopted because the C/N ratio was the most limiting optimisation parameter in previous applications of a similar optimisation problem in [41]. Scenario 2 was designed with both linear and nonlinear distance modifications (Scenarios 2a and 2b, respectively) to discuss the effect of distance distribution, as pointed out in [41]. Note that trimming tests were carried out only for GA and PSO using the baseline scenario, assuming that trimmed parameters would suffice for simulation of other synthetic scenarios similar to the baseline scenario.

The optimisation results are presented as a sequence of contributions from all the generators to each anaerobic digester. This optimised contribution sequence can be considered a suggested logistic plan for the co-substrate distribution as follows: once enough substrate has been produced and stocked on a waste generator, a truck of 20 metric tonnes capacity would be fully loaded with substrate from the corresponding waste generator, disregarding the truck waiting time before starting each route; once fully loaded, the truck is assumed to travel to the waste receptor without further stops (assuming it always follows the same route). As long as the cycle of supply routes of all involved waste generators is completed within the AnD retention time of 20 days, the properties of the resulting blending should not vary significantly, especially considering that every waste receptor would have a receiving system for these external organic substrates, where they would be stored and blended before being added to the AnD system. The specific start and finish time for each route along the day have not been considered; this does not affect the optimisation, although it has been noted that it has considerable impact on real-world implementation.

### 3.3. Algorithm Performance Comparison and Scenario Analysis

The simulation results for each scenario are shown in Table 5. For every scenario and for ACO, GA and PSO, this table shows the best cost index (B) achieved, elapsed optimisation time, and additional parameters related to the performance of the AnD systems: total biogas production, average organic load, average carbon/nitrogen ratio and average alkalinity.

In the baseline scenario, ACO and GA show higher biogas production than PSO (23% and 30% higher, respectively). However, they show a slightly lower *B* index achieved (4% and 11% lower). This result indicates that although one of the main goals of AnD optimisation involves maximising biogas production, it is not all that matters because there are other parameters also subjected to optimisation. PSO appears to find a solution with lower biogas production but better optimises other quality parameters, such as the C/N ratio and alkalinity. However, it is remarkable that out of the three algorithms, ACO and GA find a “similar solution” (prioritising high biogas production), and PSO fins a significantly different solution (prioritising other quality-related parameters).

In Scenario 1, the COD concentration was increased tenfold. This was done to compare the efficiency of algorithms to optimise substrates with high organic loads, which is especially meaningful for maximisation of biogas production. For this scenario, ACO and GA have better performance, according to the best index *B* achieved. As a natural consequence of substantial COD increases, biogas production also dramatically increases. However, PSO is unable to achieve a competitive solution in relation to both ACO and GA within this scenario and the baseline scenario, respectively.

In Scenario 2a, a tenfold lineal increase in the geographical distances between facilities was conducted. This scenario allows comparing how well each algorithm can handle situations where most substrates have long distances. For this scenario, ACO is unable to find a feasible solution. On the other hand, GA and PSO find a solution, but the corresponding biogas production is far lower than that obtained in the baseline scenario (48% and 41% lower biogas production for GA and PSO, respectively).

Alternatively, Scenario 2b shows the optimisation results when nonlinearly modifying geographical distances between facilities by the square root of the original distances. This modification allows understanding which algorithm would be more favoured by a more equally distributed geographic location of plants. All the algorithms tested are able to find a feasible solution, showing that GA has the best performance (both in terms of best *B* and biogas production). On the other hand, ACO shows the worst best index *B* achieved.

Scenario 3 was modified by a threefold increase in available volume from all sources. The presented modification allows studying the performance of each algorithm when the total number of possible solutions is much greater. ACO shows noticeably poor performance, below the best index *B* achieved by ACO in former scenarios. Although GA shows better performance than PSO in terms of the best index *B* achieved, biogas production appears similar to that in other scenarios.

In Scenario 4, an increase in the C/N ratio for waste generators W1–W12 was conducted. This modification would test the ability of each algorithm when one of the restrictions (i.e., C/N ratio) requires more adjustments. In this case, PSO shows the best index *B* achieved, although it presents the lowest biogas production. Similar to the baseline scenario, ACO shows slightly better performance than GA in terms of the best *B* achieved, but GA still has slightly better biogas production.

Geographical distance modification was performed with two alternative scenarios. Scenario 2a includes a lineal modification of the distance matrix (tenfold), and Scenario 2b considers a distance modified by the square root of the original distance. The relative locations of all involved waste generators and receptors in the case study are shown in Figure 4. For the baseline scenario, waste generators are homogeneously geographically distributed, but receptors are located in a relatively small area—i.e., the geographical distance difference of each receptor from the emitters might be negligible by the optimisation—which may be interpreted as a single receptor with higher volume capacity, caused by the geographical overlapping of waste receptors, or the “big dot” effect. The linear modification of geographical distances in Scenario 2a does not alter this relative distribution, but the nonlinear modification in Scenario 2b does so, avoiding this “big dot” effect by dispersing Receptors A, B, and C in the geographical space. It is important to note that for Scenario 2a, ACO was unable to find a viable solution, and both GA and PSO achieved a relatively poor solution compared to the corresponding solutions for the baseline scenario.

Figure 5 shows the resulting blending profile for the baseline scenario and Scenario 2b. For the baseline scenario, PSO tends to balance the blending of substrates with a low organic load content—i.e., from W1–W12—and selects noticeably lower amounts of high organic load substrates—i.e., from C1–C7— than ACO or GA. This observed behaviour is similar between the three waste receptors A, B and C. On the other hand, ACO and GA tend towards selective blending, showing similar preferences for both receptors B and C. For receptor A, the GA algorithm tends towards slightly more homogeneous blending. In any case, both ACO and GA include more substrates of high organic load—i.e., from C1–C7—except for receptor C.

For Scenario 2b, the ACO blending profiles are similar to those obtained in the baseline scenario—showing a certain tendency to include particular waste generators, —although varying the substrates that the algorithm selects. The GA blending profiles obtained in Scenario 2b are the most affected by geographical distance distortion. GA appears to balance the blending of substrates from all waste generators, much like PSO for both scenarios 2a and 2b. Additionally, the GA blending profile for Scenario 2b accounts for more industrial, high organic load wastes—i.e., from C1–C7—than the PSO blending profile, which remains relatively similar between the baseline scenario and Scenario 2b.

## 4. Discussion

Simulations with the optimisation algorithms ACO, GA and PSO were performed, showing successful optimisation results in almost all scenarios. Data from a real case study were used to carry simulations of centralised anaerobic co-digestion blending. As detailed in Section 3, these datasets are composed of 19 organic waste generators and three organic waste receptors within the context of a sanitation network in an area of high industrial activity in Catalonia. This case study composes the baseline scenario. In that previous work, the potential impacts of optimising AnD with wastes from external sources were already demonstrated, bearing up to 77% cost savings regarding waste management. Different modifications were made to this dataset to compare the performance of the ACO, GA, and PSO algorithms under different conditions to assess the performance of each optimisation algorithm in relevant situations. Regarding the optimisation problem, the C/N ratio is the dominant restriction, as was previously seen in [41]. This is the reason why this parameter is included in the discussion of the results, together with biogas production.

For the baseline scenario as seen in Table 5, PSO shows the best index *B* achieved but also the lowest biogas production. However, PSO also shows the lowest C/N ratio, which might play a role in achieving the best solution, compensating for the lack of biogas produced. If biogas production is increased by the design of a particular setup, this could be trimmed by the corresponding weight in the objective function *B* as a trade-off among the different parameters involved. The results obtained have been considered convenient for the installation under study and improved dramatically performance obtained in the baseline scenario [41]. However, both ACO and GA generally show higher amounts of biogas production, but their best index *B* values achieved are below that of PSO, and their C/N ratios are above 50. 

As shown in Figure 5, ACO and GA show similar behaviours for the baseline scenario, prioritising specific substrates. A first hypothesis suggests that prioritised substrates would be those with higher COD since they would allow higher biogas production. On the other hand, PSO shows a different strategy blending more available substrates and tends to exclude industrial substrates. This trend may point to PSO performing a conservative strategy where it is avoided in all cost situations where the operation of the AnD would be put at risk. Therefore, the general trend is that ACO and GA solve the presented optimisation problem by maximising biogas production and pushing restrictions to the limit, while PSO tends to balance biogas maximisation and the C/N ratio trade-off. In addition, note that PSO has the shortest execution times and ACO the largest, which is observed for all scenarios, indicating PSO to be more computationally efficient, where even here, the execution time is not a drawback for real implementation with the values obtained.

The similarities between ACO and GA and the differences between those and PSO could be partially explained by the nature of these algorithms. Both ACO and GA tend to explore the search space of solutions around the borders, thus increasing the number of non-feasible solutions but also increasing the chances of finding a “rare” solution with a higher best index [72,73]. Thus, these algorithms appear to be based on relatively independent behaviour between particles so that each one can explore separate areas of the border search space and be able to find different non-redundant solutions. On the other hand, PSO algorithm exploration of the search space is based on dependent behaviour between neighbouring particles, which does not encourage particles to explore the limits of the search space. Instead, it promotes the exploration of other mid-term areas between the centre and the borders of the search space. This could help explain why the PSO algorithm attains solutions within shorter execution times but also with generally lower biogas production. Hence, PSO would tend to be a conservative strategy where instead of selecting the most promising solution, single ant or particle, it would prioritise a consensus between the best neighbourhoods.

As detailed in Section 3, Scenario 1 is modified by increasing the organic load of all substrates tenfold. Thus, the dominant condition, in this case, is that organic waste valorisation is fostered, leading to higher biogas production. As observed in Table 5, ACO and GA show better performance than PSO in this scenario, but GA is more efficient since its attained biogas production is noticeably higher than that achieved with ACO.

Additionally, as detailed in Section 3, Scenarios 2a and 2b include a geographical location modification of the involved facilities. As observed in Figure 4, the relative distances between waste generators and receptors (R1, R2, R3) are not modified by lineal modification of the distance when the map plot of the baseline scenario is compared to that of Scenario 2a. However, nonlinear modification of geographical distances in Scenario 2b leads to a different map plot, where waste receptors are more dispersed between them in relation to waste generators (Figure 4). The effect of this distortion of distances is that waste receptors are more separated, thus avoiding geographical overlapping of waste receptors, or the “big dot” effect, i.e., assimilating closer plants as a single centralised plant from a geographical perspective.

Hence, Scenario 2a is modified by tenfold increasing the geographical distances between waste generators and receptors, making geographical distance a dominant condition for optimisation. In this case, ACO is unable to find a solution, and both GA and PSO show extremely poor performance when compared with the baseline scenario, as shown in Table 5. The linear modification of distances of Scenario 2a shows the performance of each algorithm under the pressure of cases with high geographical distances. This pressure case of Scenario 2a was especially relevant to test because it can significantly impact the logistics processes. On the other hand, Scenario 2b presents a different trend due to the nonlinear modification of distances. As detailed in Table 5, GA shows the best performance, and PSO shows better performance than ACO even in terms of biogas production.

As observed in Figure 5, ACO and PSO maintain similar blending profiles, while GA and PSO also exhibit similar blending profiles, but including GA results in a greater volume of industrial substrates. This shared behaviour between GA and PSO is exclusive to Scenario 2b, but it might indicate that GA behaves similarly to PSO in this case. However, from the operational point of view, GA solutions involve major risks since they tend to include more industrial substrate than PSO solutions.

In Scenario 3, a threefold increase in the volume of all waste generated was performed. First, this result implies that the search space—i.e., the total number of combinations and possible solutions—drastically increases. In this case, Table 5 confirms a similar trend observed for previous scenarios, where GA obtained the best performance and ACO the worst. Again, this finding is consistent with the observation that the ACO algorithm attains weaker performance than GA and PSO for this particular case and that GA and PSO attain similar performance in this study, although GA appears to generally provide better performance than PSO.

Finally, Scenario 4 was composed of increasing the C/N ratio of W1–W12 substrates. These substrates originally conformed to sewage sludge with a low nitrogen load, but in Scenario 4, the drastic increase in the C/N content of sewage sludge was the dominant condition to be tested. Table 5 also presents a summary of the results for Scenario 4, where PSO shows the best performance and GA the worst. The main observation is that the PSO algorithm is more able than the GA to manage situations with high nitrogen loads or major restrictions, while the GA has more potential to maximise biogas production. However, it is more sensitive to high nitrogen loads because it reduces available space to acquire industrial wastes with both high organic loads and high nitrogen loads.

The developed algorithms have successfully optimised the AnD network of the case study, and their performances have been tested under different conditions (i.e., Scenarios 1–4). Simultaneous logistics and quality optimisation of a network of existing waste management facilities is a gap in the current state of the art due to its ad-hoc nature and its interdisciplinarity: there are specialized works for logistics optimisation such as in [45,46], but they do not include process optimisation. The present study implements this logistic optimisation by minimising a cost function designed to this end. The reason for choosing this approach is also based on the need for professionals who manage the AnD network considered here to have a decision support tool capable of integrating logistics and process performance optimisation. And, in addition, to have the ability to handle changing operation conditions and scenarios, as it actually happens in real facilities.

It is also worth noting that it exists a variety of sensors for the determination of physical-chemical parameters that could complement the sensor network considered in this case study, such as a variation of the ones presented in [47]. These sensors could provide additional insight, especially if combined with GIS and process optimisation, and also facilitate the real-time implementation of the presented approach. Additionally, they could also be used as control mechanisms for those cases where ACO and GA optimisation is applied, since attained optimised outcomes pushed quality restrictions of the AnD process close to their thresholds. However, there is a trade-off between information (i.e., data gathered from new sensors) and resources (e.g., implementation, maintenance) which has to be taken into account when considering new sensors. 

Overall, this study presents a step forward towards the integrated optimisation of AnD networks, making an innovative attempt to couple logistics and quality optimisation of the centralised digestion process of a real AnD network. 

## 5. Conclusions

In this study, three approaches were developed for the simultaneous optimisation of multiple AnD systems based on ACO, GA and PSO. These methods were applied to a case study based on real data from an AnD network in the area of the Besòs River basin in Catalonia. The performance of each optimisation approach was evaluated. All the approaches successfully optimised biogas production for simulated scenarios while preserving some practical restrictions in optimisation.

For the baseline scenario, ACO and GA allowed maximum biogas production by placing restrictions on the limits of safe operations. On the other hand, PSO solved the optimisation problem with a more conservative strategy where biogas production is lower than that in ACO or GA solutions, in favour of the best AnD operation conditions (i.e., by adjusting the C/N ratio and alkalinity).

In those cases with high opportunities for biogas production (i.e., Scenario 1), GA and ACO would perform the best due to their capabilities of maximising biogas production over that of PSO. GA would perform as the best optimisation algorithm both for cases where distances are significantly different amongst them (i.e., Scenario 2b) and for cases where higher volumes should be handled (i.e., Scenario 3), presumably due to GA’s computational potential. Finally, for those cases where other quality-related parameters are restrictions (i.e., Scenario 4), PSO would be the best performing algorithm.

The present study shows an innovative contribution to optimize the performance of centralized AnD systems, combining logistical and quality parameters. To the authors’ knowledge, this optimization has not yet been addressed in the literature for an AnD network. In addition, the framework has proven its effectiveness in minimizing the total distance travelled to transport the waste and maximizing biogas production. At the same time, the physical-chemical parameters of the process have been kept within their operational limits.

Further work may include methodologies to improve social impact factor quantification in the optimisation, which might allow better characterisation of the logistic impact of each substrate generator. Additionally, the development of logistic route simulations would be required to enhance real-world distribution planning considering daytime, travel frequency, dynamic waste production-consumption coupled with stocking problems and other time-related issues key to logistic planning.

## Figures and Tables

**Figure 1 sensors-22-01857-f001:**
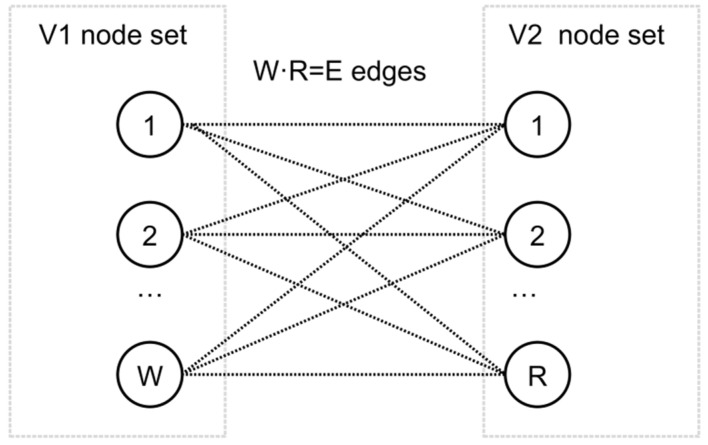
Generic representation of the posed matching problem.

**Figure 2 sensors-22-01857-f002:**
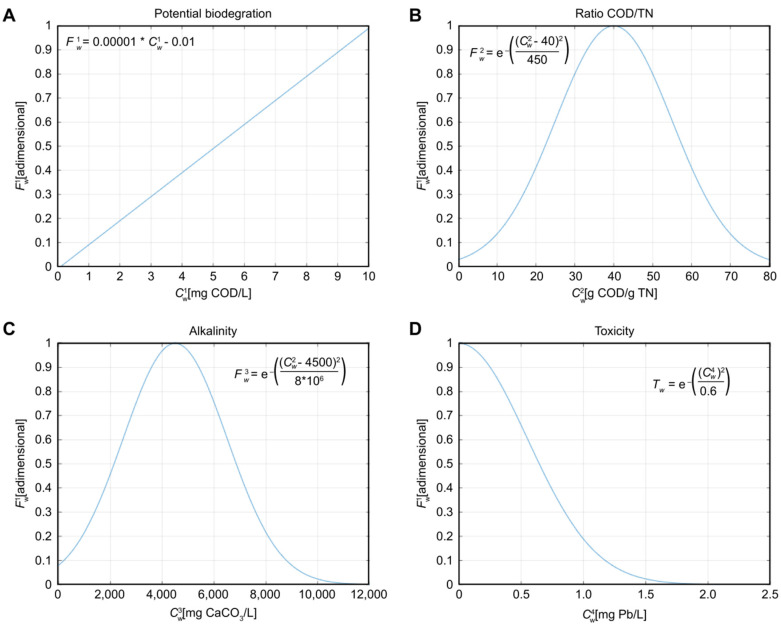
(**A**) $F_{w}^{1}$, (**B**) $F_{w}^{2}$, (**C**) $F_{w}^{3}$ and (**D**) $T_{w}$ equations used for dimensionless coefficient calculation.

**Figure 3 sensors-22-01857-f003:**
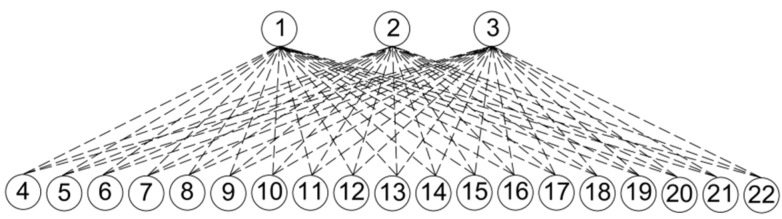
Bipartite graph of the case study.

**Figure 4 sensors-22-01857-f004:**
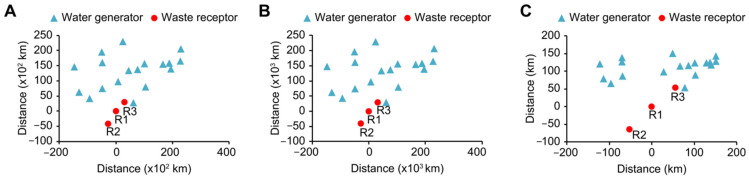
Map of waste generators and waste receptors R1–R3 for Baseline Scenario (**A**), Scenario 2a (**B**) and Scenario 2b (**C**). Distance is expressed as longitudinal distance (X-axis) and latitudinal distance (*Y*-axis) with respect to the R1 plant.

**Figure 5 sensors-22-01857-f005:**
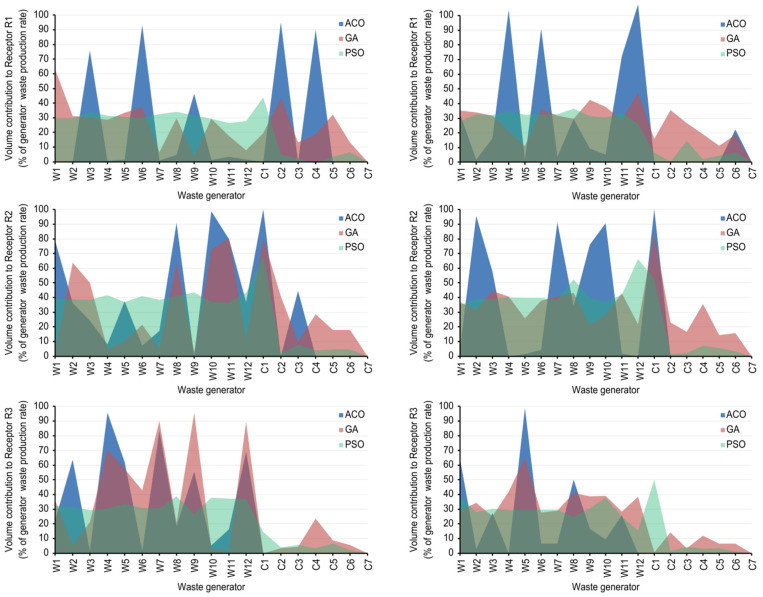
Blending profiles for every waste receptor and ACO, GA and PSO algorithms for the baseline scenario (**left**) and Scenario 2b (**right**).

**Table 1 sensors-22-01857-t001:** Summary of trimming tests for the GA.

Tested Parameters		Best Index (*B*)	Time(s)
Crossover Fraction	0.2	0.0274	525.73
	0.5	0.0295	574.46
	**0.8**	**0.0304**	**537.98**
Elite Count	**0.05**	**0.0309**	**541.48**
	0.15	0.0300	536.15
	0.3	0.0291	554.83

**Table 2 sensors-22-01857-t002:** Summary of trimming tests for PSO.

Tested Parameters		Best Index (*B*)	Time(s)
Cognitive Attraction	0.2	0.0293	82.86
	0.5	0.0304	62.93
	**0.8**	**0.0322**	**57.26**
Social Attraction	1.05	0.0301	55.78
	**1.25**	**0.0329**	**74.98**
	1.95	0.0308	63.45

**Table 3 sensors-22-01857-t003:** Waste generator dataset, including distance between waste generators and receptors and characterisation of each receptor of the case study (Baseline Scenario or Scenario 0).

Waste Generator ID	Vw (L by Day)	COD (mg/L)	C/N	Alk (mg/L)	Tw (mg/L)	R1	R2	R3
						Distance to R1 (km)	Distance to R2 (km)	Distance to R3 (km)
W1	27,600	19,900	17.8	4300	1.55	5.3	20.5	9.5
W2	47,000	16,900	20.6	3200	1.36	35.9	33.9	45.7
W3	46,300	18,600	19.4	10,100	1.42	21.8	16.6	28.9
W4	20,200	23,400	15.6	3400	1.38	30.4	43.2	19.7
W5	38,400	21,100	17.9	4500	1.35	19.7	24.7	12.4
W6	34,400	18,800	14.0	3800	1.61	14.8	19.9	15.9
W7	13,800	22,600	15.3	2700	1.57	32.1	44.9	18.4
W8	4400	22,100	15.2	1800	2.30	26.5	31.6	27.7
W9	10,800	21,700	15.1	5300	0.93	20.3	33.1	8.8
W10	9500	20,400	15.5	2500	1.28	30	24.8	37.1
W11	17,000	23,300	14.8	7800	0.98	36.9	31.7	44
W12	6500	20,100	16.5	3100	1.40	20.5	33.3	8.7
C1	9000	667,400	42.5	250	0.01	15.9	11.1	23
C2	9000	497,400	461.8	330	0.01	7	12	17.9
C3	9000	155,900	3118.1	60	0.02	27.9	40.7	17.2
C4	9000	459,100	274.1	660	0.10	16.2	11.1	22.9
C5	9000	657,200	2330.6	630	0.01	52.8	65.6	43.8
C6	9000	266,200	2832.4	20	0.01	56.1	33	21
C7	9000	262,100	32,768.4	110	0.01	36.7	24.1	66.4
Maximum Volume (L/day)						122,000	146,000	111,000
COD (mg/L)						18,600	19,100	18,200
C/N						19.1	20.3	18.4
Alk (mg/L)						3100	2900	3400
Tw (mg/L)						1.41	1.68	1.53

**Table 4 sensors-22-01857-t004:** Synthetic scenarios created from original Scenario 0 in Table 3. Description of data alteration procedure.

ID	Description
Baseline (Scenario 0) scenario	Scenario based on data form real case study (see Table 3)
Scenario 1	High COD (×10 COD concentration)
Scenario 2a	Linear modification of distances: ×10 distances
Scenario 2b	Nonlinear modification of distances: square root of original distance
Scenario 3	High volumes (×3 volumes)
Scenario 4	C/N variations (increase of W1–W12 C/N ratio to the 50–60 range)

**Table 5 sensors-22-01857-t005:** Summary of algorithm performance. The best value *B* is highlighted. Scenario 2a feasible results (*) are associated with a poor solution, so no direct comparison is conducted.

**Scenario**	**Baseline Scenario**			**Scenario 1**		
**Optimisation Method**	**ACO**	**GA**	**PSO**	**ACO**	**GA**	**PSO**
Best Index (B)	0.0336	0.0313	**0.0349**	**0.0330**	0.0328	0.0211
Time (seconds)	595.46	325.37	**90.20**	1825.34	1035.53	**221.74**
Total Biogas Production (Nm^3^/d)	25,657	**27,133**	20,852	114,870	**198,284**	102,927
Avg Organic Load (kg COD/m^3^·d)	2.32	**2.59**	2.09	9.67	**17.46**	9.73
Avg C/N ratio (limited below 60)	50.5	56.1	**46.4**	24.1	54	**32.4**
Avg Alkalinity (g CaCO_3_/m^3^)	3079	3141	**3245**	3183	3193	**3282**
**Scenario**	**Scenario 2a**			**Scenario 2b**		
**Optimisation Method**	**ACO**	**GA**	**PSO**	**ACO**	**GA**	**PSO**
Best Index (B)	-	0.0001 *	0.0001 *	0.0287	**0.0333**	0.0319
Time (seconds)	-	62	60	669.56	193	**63.61**
Total Biogas Production (Nm^3^/d)	-	14,278	12,325	17,468	**25,404**	19,237
Avg Organic Load (kg COD/m^3^·d)	-	1.6	1.4	1.69	**2.5**	2.24
Avg C/N ratio (limited below 60)	-	45.9	32.8	23.4	55.2	**32.4**
Avg Alkalinity (g CaCO_3_/m^3^)	-	3298	3319	**3338**	3174	3221
**Scenario**	**Scenario 3**			**Scenario 4**		
**Optimisation Method**	**ACO**	**GA**	**PSO**	**ACO**	**GA**	**PSO**
Best Index (B)	0.0077	**0.0324**	0.0300	0.0339	0.0319	**0.0354**
Time (seconds)	671.03	548.46	**68.10**	1824.94	350.89	**86.47**
Total Biogas Production (Nm^3^/d)	17,224	33,657	**35,395**	20,770	**23,326**	19,524
Avg Organic Load (kg COD/m^3^·d)	1.70	2.79	**2.92**	2.02	**2.23**	1.90
Avg C/N ratio (limited below 60)	19.1	26.3	**32.9**	35.9	56.4	**37.8**
Avg Alkalinity (g CaCO_3_/m^3^) (limited above 2500)	2985	3020	**3233**	3217	3199	**3272**

## Data Availability

No new data were created or analyzed in this study. Data sharing is not applicable to this article.

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
