# Peer review of "Comparison of Optimisation Algorithms for Centralised Anaerobic Co-Digestion in a Real River Basin Case Study in Catalonia"

_sensors, 2022, doi:10.3390/s22051857_

Round 1

Reviewer 1 Report

This paper addresses an interesting and important point, regarding finding the most efficient balance between substrate emitters and AnD receptors, which is beneficial to waste management and biogas energy production.

The paper is well written with a well designed methodology. I would recommend its publication in the journal "Sensors" upon addressing the following point:

  • The audience of "Sensors" would definitely be interested to know how novel real-time remote sensors could benefit data collection which could feed the presented algorithms. Would be nice to add a paragraph with more details on data collection techniques and also bottlenecks regarding data collection. 

Author Response

(Please see the attachement to view changes made in manuscript)

R1C1: The audience of "Sensors" would definitely be interested to know how novel real-time remote sensors could benefit data collection which could feed the presented algorithms. Would be nice to add a paragraph with more details on data collection techniques and also bottlenecks regarding data collection. 

Response 1: New text has been added in the final part of the Discussion section, where more detail is added regarding sensor implementation and the potential contribution to the developed optimization framework.

Reviewer 2 Report

Dear Authors,

The aim of this paper is to compare optimization algorithms for centralized anaerobic co-digestion in a real river basin case study in Catalonia. This study should be supplemented by the following:

  1. The abstract must be rewritten. It does not clearly present: the current situation, the purpose of the study, the methodology, the main results, and the conclusions of the study. The abstract must be completed with the main results obtained.
  2. To emphasize the need for this study.
  3. The introductory section can be completed with other studies conducted in this field.
  4. To highlight the objectives of the study.
  5. To highlight the gaps filled by the present study.
  6. The conclusions section should be completed with a review of the study.

Author Response

(Please see the attachement to view changes made in manuscript)

R2C1:    The abstract must be rewritten. It does not clearly present: the current situation, the purpose of the study, the methodology, the main results, and the conclusions of the study. The abstract must be completed with the main results obtained.

Response 1: The Abstract has been rewritten with more mentions to the current situation of AnD networks, the methodology that allows a simultaneous logistic and physical-chemical optimization, and a highlight of the results obtained by each algorithm.

R2C2:    To emphasize the need for this study.

Response 2: A new paragraph in the Introduction section has been developed highlighting the existing gaps that the present work can fulfill, in order to emphasize the need for this study.

R2C3:    The introductory section can be completed with other studies conducted in this field.

Response 3: A new paragraph in the Introduction section has been developed detailing current trends in the field of the present study.

R2C4:    To highlight the objectives of the study. 

Response 4: A new paragraph in the Introduction section has been added detailing the objectives of the study.

R2C5:    To highlight the gaps filled by the present study.

Response 5: New text has been added to the final part of the Discussion section to highlight the gaps filled with the study, and to compare its contribution with previous studies in the field.

R2C6:    The conclusions section should be completed with a review of the study.

Response 6: A new paragraph has been added to the Conclusions section summarizing the main contributions and innovations of the present study.

Reviewer 3 Report

The nature and scope of this paper are relevant to the conversion of organic waste to energy carriers such as biogas. It is positive that in this paper, three evolutionary optimisation algorithms, namely, ant colony optimisation, genetic algorithms and particle swarm optimisation, are compared to address the optimisation of this process. The needs analysis is reasonably developed, even with a somewhat limited explanation of its conduct in relevant case studies, policy documents, academic publications, and references provided. However, there is no systematic comparison between the solutions proposed by the authors and existing solutions. Various methods are outlined in the paper approach, but there is no sound justification for why the research is novel and innovative. The specific paper's objectives are generous but should be more clearly formulated. As outlined, the expected results are appropriate to the general need for adaptation and use of technologies related to the circular economy, but the authors' innovative contribution should be better explained.

Author Response

(Please see the attachment to view changes in manuscript)

R3C1:    The needs analysis is reasonably developed, even with a somewhat limited explanation of its conduct in relevant case studies, policy documents, academic publications, and references provided.   However, there is no systematic comparison between the solutions proposed by the authors and existing solutions. Various methods are outlined in the paper approach, but there is no sound justification for why the research is novel and innovative.

Response 1: New text has been added in the Introduction and Discussion sections, highlighting the potential gaps of previous works in the field, and comparing their contributions with the ones provided by this paper, to highlight the novelty and innovation of this research.

R3C2:    The specific paper's objectives are generous but should be more clearly formulated .

Response 2: A new paragraph has been added in the Introduction section to clearly state the objectives of this study.

Reviewer 4 Report

Dear Authors, 

This is a very well-written paper. It could be included in the paper mentioned chemical sensors, as could be reason to detection limit and operational condition as well. 

The only issue I am having with is that, I was thinking, the current manuscript would have more at least some connectors to the need of chemical detections limits/methods/thresholds of oxygen, nitrogen, sulfur.  

Author Response

(Please see the attachment to view changes in manuscript)

R4C1:    The only issue I am having with is that, I was thinking, the current manuscript would have more at least some connectors to the need of chemical detections limits/methods/thresholds of oxygen, nitrogen, sulfur.  

Response 1: New text has been added to the final part of the Discussion section, where more detail has been added regarding the risks of chemical restriction thresholds, and how real-time sensors used as control mechanisms could help minimizing such risks.

Round 2

Reviewer 2 Report

Dear Authors,

I accept the current form.

Reviewer.

Reviewer 3 Report

The paper could be accepted in the present form.